# The field and resistance training loads of academy rugby league players during a pre-season: Comparisons across playing positions

David Anthony Moore[1,2]*[¤], Ben Jones[1,2,3,4,5][☉], Jonathon Weakley[1,6,7][☉], Sarah Whitehead[1,2,8][☉], Kevin Till[1,2][☉]

**1** Carnegie Applied Rugby Research (CARR) Centre, Carnegie School of Sport, Leeds Beckett University, Leeds, United Kingdom, **2** Leeds Rhinos Rugby League Club, Leeds, United Kingdom, **3** England Performance Unit, Rugby Football League, Leeds, United Kingdom, **4** Division of Exercise Science and Sports Medicine, Department of Human Biology, Faculty of Health Sciences, the University of Cape Town and the Sports Science Institute of South Africa, Cape Town, South Africa, **5** School of Science and Technology, University of New England, Armidale, New South Wales, Australia, **6** Sports Performance, Recovery, Injury, and New Technologies (SPRINT) Research Centre, Australian Catholic University, Brisbane, Australia, **7** School of Behavioural and Health Sciences, Australian Catholic University, Brisbane, Australia, **8** Leeds Rhinos Netball, Leeds, United Kingdom

☉ These authors contributed equally to this work.
¤ Current address: Carnegie School of Sport, Leeds Beckett University, Headingley Campus, Leeds, United Kingdom
* A.Moore@leedsbeckttt.ac.uk

**Data Availability Statement:** Dataset is available from the Leeds Beckett Repository https://doi.org/10.25448/lbu.20075939.v1.

## Abstract

Male academy rugby league players are required to undertake field and resistance training to develop the technical, tactical and physical qualities important for success in the sport. However, limited research is available exploring the training load of academy rugby league players. Therefore, the purpose of this study was to quantify the field and resistance training loads of academy rugby league players during a pre-season period and compare training loads between playing positions (i.e., forwards vs. backs). Field and resistance training load data from 28 adolescent male (age 17 ± 1 years) rugby league players were retrospectively analysed following a 13-week pre-season training period (85 total training observations; 45 field sessions and 40 resistance training sessions). Global positioning system microtechnology, and estimated repetition volume was used to quantify external training load, and session rating of perceived exertion (sRPE) was used to quantify internal training load. Positional differences (forwards n = 13 and backs n = 15) in training load were established using a linear mixed effect model. Mean weekly training frequency was 7 ± 2 with duration totaling 324 ± 137 minutes, and a mean sRPE of 1562 ± 678 arbitrary units (AU). Backs covered more high-speed distance than forwards in weeks two (p = 0.024), and 11 (p = 0.028). Compared to the forwards, backs completed more lower body resistance training volume in week one (p = 0.02), more upper body volume in week three (p< 0.001) and week 12 (p = 0.005). The findings provide novel data on the field and resistance-based training load undertaken by academy rugby league players across a pre-season period, highlighting relative uniformity between playing positions. Quantifying training load can support objective

**Funding:** The author(s) received no specific funding for this work.

**Competing interests:** The authors have declared that no competing interests exist.

decision making for the prescription and manipulation of future training, ultimately aiming to maximise training within development pathways.

## Introduction

Academy athletes within sport engage in structured training programmes to maximise their development and progress to senior professional levels [1]. The training dose, defined by the frequency, volume and intensity of exercise, determines the type and magnitude of the training response [2, 3]. The prescription of training by coaches remains largely instinctive, consequently research is needed to further understand training programme design [2, 4, 5]. The monitoring of training has become common practice in elite sport, with the primary purpose being to guide and inform training prescription [6], which is typically quantified with reference to frequency, intensity, time, and type of each session [7]. Practitioners have adopted the term 'training load', further classifying external training load (e.g., distance covered measured by global positioning systems; GPS) and internal training load (e.g., athlete perception of training intensity measured by session rating of perceived exertion; sRPE) to quantify exercise [8].

Microtechnology (i.e., GPS and micro-electro mechanical systems) has allowed for the detection of increasingly detailed information on external training load (e.g., tackles), which has led to widespread adoption of microtechnology to quantify training in collision based sports, including rugby league [9–11]. The intermittent movement, collision, and skill components of rugby league match play require players to have a wide range of technical (e.g., passing, kicking, tackling) and physical (e.g., strength, speed, repeated effort ability) capacities [12]. Training programmes should expose players to the specific intensity and volume of match-play throughout the training week [13]. Therefore, players engage in a diverse range of training modalities (e.g., technical-tactical, resistance training, speed and conditioning) to elicit specific adaptations [12]. sRPE offers a valid and easily implemented measure of internal training load that can be used across the various training modalities [14–16]. Ultimately, a coordinated and systematic approach quantifying both external and internal training load would provide more comprehensive insights into the overall demands of training, and support the development of specific training that aims to develop both physical and technical qualities [16, 17]. To date however, there remains limited training data available within rugby league, at any level, to use as a reference for what may be considered an appropriate load [18].

Research in rugby league has presented the characteristics of match-play [19–21] the physical qualities of athletes across playing standards [22, 23] and evaluated training interventions [11], but few studies have quantified training load. Gabbett [24] was the first to describe the typical training activities used to prepare professional players for competition and demonstrated the demands of competitive play were not being well matched to those observed in training. Black et al. [18] then quantified the field-based external training load of professional rugby league players using GPS across the playing season, but did not account for resistance training or internal load. Daniels et al. [11] reported on strength, power, and endurance characteristics and their association with training load during a 7-week pre-season. In doing so, Daniels et al. [11] reported the distribution of multimodal rugby league training, where field-based sessions totalling 48.5%, gym-based 37.9%, and wrestle sessions 13.6% over the pre-season period. While these studies expand our knowledge surrounding training, the research is exclusive to the adult professional standard with limited information available in academy players. Dobbin et al. [25] examined the influence of training load on body composition over a 14-week pre-season in academy rugby league, using sRPE as the universal measure of training

load. Most recently, McCormack et al. [26] reported weekly training volumes ranging between 359 and 1,033 mins, depending on the phase of the season. However, quantification of volume was an estimation by coaches of a 'typical week' and did not provide objective data on the accumulated weekly training load in academy rugby league. Therefore, more detailed analyses combining microtechnology and sRPE to quantify training load across modalities (i.e., field and resistance training) is warranted.

As both excessive and insufficient training load may impede athletic development, understanding the specific training loads undertaken by academy rugby league players is important for the planning of future training to maximise athletic performance, injury prevention, playing progression, and general wellbeing [3]. Differences in training load, alongside differences in certain physical capabilities associated with specific playing positions and standards, may influence one's understanding of appropriate training prescription [1, 27]. No study has quantified an academy rugby league pre-season period, considering positional demands for both field- and resistance-based training loads. A greater understanding of academy rugby league training will help practitioners plan, deliver, and evaluate training. Therefore, the aim of the current study was to quantify the field- and resistance-based training loads of an academy (U19) rugby league pre-season period and compare these training loads by positional groups across training weeks.

## Methods

### Study design

A retrospective observational study design was used to quantify the field- and resistance-based training load of academy-level rugby league players from one professional club during a pre-season period. The study included field- and resistance-based training during the 2019/2020 pre-season, representing a 13-week period from November to February. The pre-season was comprised of four separate periodised training blocks (block one: two weeks [nine training days], block two: three weeks [fifteen training days], block three: three weeks [six training days], and block four: five weeks and [sixteen training days], post winter break) focusing on rugby specific skills, aerobic and anaerobic conditioning, sprinting, and muscular strength. Block three included a 11-day rest period (for the winter break). The data set included training loads being recorded via microtechnology units, the resistance-based repetition method (sets and reps) and sRPE. All training took place at the training ground of the professional club supervised by qualified coaching staff.

### Participants

Twenty-eight male academy rugby league players from a single professional club playing in the Under-19s Super League competition (age 17 ± 1 years, body mass 90.1 ± 13.2 kg, height 178.6 ± 6.4 cm) participated in the study. Due to the applied nature of this research, the number of participants included was dictated by the resource constraints of the professional sporting organisation [28]. A total of 46 training days were captured over the 13-week pre-season period, with a mean ± standard deviation (SD) of 37 ± 7 field-based and 33 ± 7 gym-based training observations per player. Players were split into two positional groups to provide position-specific findings, forwards (n = 13) and backs (n = 15). All training data collected was part of regular practice, training content was not in any way influenced by the research. The study received approval by the Leeds Beckett University Ethics Advisory Committee, and all participants and parents / guardians (where needed) provided written consent.

## Procedures

Players underwent 13-weeks of training, consisting of strength and conditioning and rugby league specific skills training. To capture the concurrent training employed during the pre-season period, training was categorised into field and gym sessions. Total weekly training load were summated and presented to make comparisons between weeks, alongside mean sessional training load representing the within week average. The field sessions represented training sessions including technical/tactical skills, aerobic and anaerobic match based conditioning sessions, speed sessions, and combat sessions similar to previous reports [11, 29]. Gym sessions included all resistance training undertaken during the period.

**Quantification of training load.** *Internal load*. Given the concurrent nature of the training performed (i.e., field-based and gym-based), sRPE [14, 15] was used as a consistent measure of training load across training modalities [6]. sRPE was calculated as session duration multiplied by the individual's RPE. Participants scored individual sessions using a modified Borg Category Ratio-10 RPE scale [14, 15]. All players were familiarised with the scale prior to commencement of the study. To minimise bias from the most recent phase of exercise, participants recorded their RPE approximately 30 min post-training. Recordings were taken non-verbally with each participant on their own and blinded from other scores to control for external influences. Mean sessional, weekly, and total sRPE-TL for the 13-week period was calculated for both gym and field.

*External load*. Microtechnology units (Optimeye S5, Catapult Innovations Melbourne, Victoria) were used to quantify the field-based locomotive loads. The validity and reliability of microtechnology devices measuring instantaneous velocity, collision count, and accelerometer derived PlayerLoad[TM] has been established [30, 31]. Players wore an undergarment which housed the microtechnology unit positioned in-between the scapulae as per manufacturer instructions. The same units were worn for repeated observations, and the devices were switched on 30-minutes prior to commencement of training [32]. Data were downloaded from the microtechnology devices using the proprietary software (Catapult Openfield, v.1.21.1). Velocity was calculated via the Dopler shift method, and the minimum effort duration was set at 1-second [31]. The instantaneous 10-Hz speed data and collision event files were exported, and all further analysis was carried out using the statistical software Jamovi (The Jamovi Project, jamovi Version 1.6.18.0).

Analysis of field-based locomotive loads included total distance (meters), high-speed running (HSR) meters with the speed threshold set at $>5$ m·s$^{-1}$ (HSR and sprinting were aggregated to represent total-HSR) [12], average speed (meters per minute), acceleration density (meters per second), PlayerLoad (arbitrary units), and tackles (count), aligned with those previously reported in rugby league [20, 33, 34]. Acceleration density was calculated as the summation of absolute acceleration and deceleration values across the duration of the training session which was divided by the total time spent on field to calculate mean acceleration to represent the relative output.

Resistance training volume were quantified using the repetition method [35]. Programmes were assigned including the exercises, number of sets, and repetitions to be completed. From these prescribed programmes, upper- and lower-body repetitions (sets multiplied by repetitions) were reported, in addition to weekly resistance training frequency and duration. The aim of the gym programme was designed to increase muscle mass and strength, while establishing fundamental movement patterns (i.e., squat, lunge, hinge, push, pull, brace and rotate).

## Statistical analysis

Data were analysed using the statistical software Jamovi (The Jamovi Project, Jamovi, Version 1.6.18.0 [Computer Software]). Descriptive statistics were calculated and presented as

means ± SD. In line with the observational approach to the study design, mixed linear modelling was used to assess the differences between positional groups and training weeks for each dependent variable (field- and gym-based variables). A mixed model was used as it can be applied to repeated-measures data from unbalanced designs, which was the case in the current study since players differed in terms of the number of training sessions they participated in. Both fixed- and mixed-effect analysis of covariance models were used to assess differential effects by weeks (weeks 1–13) of the pre-season and positional groups (forwards and backs). Week and positional group (forwards and backs) were treated as fixed effects. Random effects were associated with the individual players (subject ID). The models assessing field and gym were independent of each other. Significance was set at $p < 0.05$, and effect sizes (ES) with 95% confidence intervals were used, with effect size calculated as the reported difference divided by the pooled SD. This approach was applied to training load data to assess the pre-to-post change from the beginning to the end of the pre-season observation period. Threshold values for effect sizes were: 0.0–0.19, trivial; 0.2–0.59, small; 0.6–1.19, moderate; 1.2–2.0, large; >2.0, very large. If one or more fixed effects were statistically significant, post-hoc pairwise comparisons were performed to examine between pairs of categories of the significant factors. 95% confidence intervals (CI) of the raw and standardized coefficients were also calculated. Data are presented as an estimated marginal means ± standard error (SE), for pairwise comparisons of time periods or positional roles as 95% CI.

## Results

### Training load across the pre-season

The pre-season training loads of academy rugby league players across a pre-season period are presented in Table 1 (total weekly) and 2 (mean sessional). Average speed, and acceleration density are presented in Table 2 only, as these variables cannot be summated as weekly totals. The mean accumulated sRPE-TL for the entire pre-season phase for field and gym sessions were 10,366 ± 2,740 and 8,613 ± 1,750 AU, respectively. The mean weekly sRPE-TL for the pre-season period was 857 ± 426 and 722 ± 287 AU for field and gym, respectively. No significant differences in sRPE-TL were found between forwards and backs for either field or gym.

**Table 1. Total (Mean ± SD) weekly training load during an academy rugby league pre-season.**

| Week | Total | | | Field | | | | | | | Gym | | | | |
|---|---|---|---|---|---|---|---|---|---|---|---|---|---|---|---|
| | Frequency (n) | Duration (mins) | sRPE (AU) | Frequency (n) | Duration (mins) | sRPE (AU) | TD (m) | HSD (m) | PlayerLoad (AU) | Tackles (n) | Frequency (n) | Duration (mins) | sRPE (AU) | LB (reps) | UB (reps) |
| 1 | 8 | 441 ± 73 | 2172 ± 536 | 4 | 274 ± 80 | 1332 ± 197 | 14927 ± 2795 | 1531 ± 425 | 1543 ± 367 | 43 ± 20 | 4 | 188 ± 31 | 942 ± 188 | 940 ± 178 | 264 ± 0 |
| 2 | 9 | 485 ± 104 | 2283 ± 368 | 5 | 297 ± 84 | 1348 ± 354 | 15627 ± 4595 | 987 ± 497 | 1531 ± 520 | 36 ± 23 | 4 | 188 ± 26 | 935 ± 168 | 839 ± 184 | 264 ± 0 |
| 3 | 7 | 304 ± 26 | 1646 ± 589 | 3 | 138 ± 54 | 901 ± 346 | 10115 ± 3213 | 951 ± 439 | 1078 ± 376 | 12 ± 13 | 4 | 166 ± 50 | 775 ± 239 | 480 ± 50 | 409 ± 142 |
| 4 | 6 | 339 ± 42 | 1696 ± 275 | 3 | 167 ± 49 | 830 ± 233 | 13161 ± 2596 | 2240 ± 435 | 1341 ± 287 | 26 ± 14 | 3 | 173 ± 12 | 866 ± 142 | 583 ± 58 | 328 ± 0 |
| 5 | 8 | 402 ± 26 | 2063 ± 151 | 4 | 199 ± 44 | 1053 ± 184 | 13383 ± 1287 | 1906 ± 455 | 1464 ± 218 | 30 ± 16 | 4 | 203 ± 11 | 1010 ± 90 | 677 ± 68 | 688 ± 0 |
| 6 | 8 | 452 ± 75 | 2189 ± 358 | 4 | 232 ± 65 | 1283 ± 332 | 17233 ± 5159 | 2932 ± 974 | 1668 ± 537 | 37 ± 17 | 4 | 220 ± 0 | 907 ± 79 | 626 ± 0 | 584 ± 0 |
| 7 | 1 | 54 ± 0 | - | 1 | 54 ± 22 | - | 4440 ± 309 | 177 ± 87 | 439 ± 61 | 10 ± 8 | 0 | - | - | - | - |
| 8 | 2 | 95 ± 8 | 313 ± 61 | 1 | 36 ± 14 | 135 ± 23 | 3737 ± 148 | 381 ± 96 | 373 ± 49 | 9 ± 5 | 1 | 60 ± 0 | 196 ± 37 | 341 ± 0 | - |
| 9 | 7 | 302 ± 48 | 1134 ± 228 | 4 | 161 ± 26 | 498 ± 133 | 12521 ± 2456 | 1647 ± 513 | 1262 ± 304 | 34 ± 20 | 3 | 152 ± 12 | 661 ± 84 | 311 ± 44 | 220 ± 0 |
| 10 | 8 | 405 ± 89 | 1747 ± 313 | 4 | 234 ± 32 | 905 ± 196 | 15263 ± 2236 | 1529 ± 445 | 1629 ± 289 | 40 ± 19 | 4 | 184 ± 34 | 842 ± 205 | 298 ± 66 | 349 ± 124 |
| 11 | 8 | 333 ± 130 | 1368 ± 556 | 5 | 222 ± 46 | 938 ± 409 | 17076 ± 4947 | 1980 ± 774 | 1688 ± 572 | 56 ± 39 | 3 | 115 ± 42 | 464 ± 185 | 167 ± 0 | 391 ± 104 |
| 12 | 6 | 203 ± 71 | 891 ± 303 | 3 | 121 ± 31 | 521 ± 174 | 9490 ± 1896 | 1537 ± 735 | 1014 ± 216 | 29 ± 16 | 3 | 93 ± 41 | 405 ± 220 | 225 ± 82 | 269 ± 109 |
| 13 | 7 | 320 ± 83 | 1133 ± 338 | 4 | 193 ± 49 | 622 ± 159 | 13664 ± 2948 | 1942 ± 654 | 1410 ± 329 | 39 ± 20 | 3 | 156 ± 27 | 610 ± 113 | 175 ± 0 | 411 ± 72 |
| Mean ± SD | 7 ± 2 | 324 ± 137 | 1562 ± 679 | 3 ± 1 | 170 ± 87 | 857 ± 426 | 12601 ± 4890 | 1550 ± 888 | 1287 ± 516 | 31 ± 23 | 3 ± 1 | 159 ± 53 | 722 ± 287 | 485 ± 265 | 383 ± 137 |

*Frequency (n), Duration (minutes), RPE (AU), sRPE (AU), Player Load (AU), Tackles (n), Lower Body (LB) Repetitions (reps), Upper Body (UB) Repetitions (reps)

**Table 2. Mean (Mean ± SD) sessional training load during an academy rugby league pre-season.**

| Week | Total | | Field | | | | | | | | | Gym | | | | |
|---|---|---|---|---|---|---|---|---|---|---|---|---|---|---|---|---|
| | Duration (mins) | sRPE (AU) | Duration (mins) | RPE (AU) | sRPE (AU) | TD (m) | HSD (m) | Avg.Speed (m·min⁻¹) | Accel Density (m·s) | Player Load (AU) | Tackles (n) | Duration (mins) | RPE (AU) | sRPE (AU) | LB (reps) | UB (reps) |
| 1 | 127 ± 25 | 634 ± 173 | 79 ± 13 | 5 ± 1 | 390 ± 110 | 4316 ± 1068 | 443 ± 178 | 56 ± 9 | 0.412 ± 0.15 | 446 ± 104 | 12 ± 8 | 54 ± 5 | 5 ± 1 | 275 ± 54 | 382 ± 26 | 264 ± 0 |
| 2 | 102 ± 41 | 479 ± 244 | 65 ± 19 | 4 ± 2 | 294 ± 169 | 3660 ± 970 | 231 ± 151 | 60.4 ± 13 | 0.689 ± 0.63 | 362 ± 118 | 9 ± 10 | 60 ± 1 | 5 ± 1 | 296 ± 62 | 390 ± 25 | 264 ± 0 |
| 3 | 92 ± 44 | 504 ± 269 | 62 ± 13 | 7 ± 3 | 404 ± 101 | 5596 ± 395 | 526 ± 311 | 94.4 ± 9 | 1.59 ± 0.53 | 483 ± 125 | 5 ± 7 | 51 ± 14 | 4 ± 1 | 231 ± 82 | 245 ± 1 | 280 ± 0 |
| 4 | 115 ± 20 | 588 ± 173 | 61 ± 13 | 5 ± 2 | 318 ± 112 | 4700 ± 1051 | 800 ± 501 | 81.7 ± 17 | 1.97 ± 1.35 | 479 ± 113 | 9 ± 6 | 58 ± 2 | 5 ± 1 | 300 ± 67 | 297 ± 1 | 328 ± 0 |
| 5 | 101 ± 20 | 521 ± 125 | 52 ± 14 | 5 ± 1 | 279 ± 88 | 4519 ± 1441 | 643 ± 530 | 82.2 ± 26 | 2.71 ± 1.97 | 381 ± 172 | 8 ± 6 | 51 ± 4 | 5 ± 1 | 255 ± 41 | 345 ± 1 | 344 ± 16 |
| 6 | 113 ± 30 | 547 ± 204 | 70 ± 12 | 6 ± 1 | 402 ± 113 | 5398 ± 821 | 919 ± 558 | 78.0 ± 9 | 0.398 ± 0.05 | 522 ± 99 | 12 ± 7 | 55 ± 4 | 4 ± 1 | 227 ± 44 | 313 ± 18 | 292 ± 12 |
| 7 | 54 ± 0 | - | 54 ± 0 | - | - | 4440 ± 309 | 177 ± 87 | 82.1 ± 5 | 0.452 ± 0.10 | 439 ± 61 | 10 ± 8 | - | - | - | - | - |
| 8 | 95 ± 8 | 313 ± 61 | 36 ± 0 | 4 ± 1 | 135 ± 23 | 3737 ± 148 | 381 ± 96 | 102 ± 4 | 3.59 ± 0.15 | 373 ± 49 | 9 ± 5 | 60 ± 0 | 3 ± 1 | 196 ± 37 | 341 ± 0 | - |
| 9 | 81 ± 28 | 398 ± 127 | 45 ± 9 | 4 ± 1 | 201 ± 60 | 3471 ± 965 | 456 ± 394 | 89.6 ± 25 | 2.37 ± 1.49 | 350 ± 94 | 9 ± 8 | 52 ± 5 | 4 ± 1 | 226 ± 66 | 162 ± 1 | 220 ± 0 |
| 10 | 104 ± 26 | 463 ± 191 | 61 ± 10 | 4 ± 2 | 264 ± 118 | 3994 ± 1450 | 400 ± 386 | 70.3 ± 23 | 1.60 ± 1.25 | 426 ± 117 | 10 ± 8 | 54 ± 2 | 5 ± 1 | 246 ± 72 | 165 ± 1 | 216 ± 4 |
| 11 | 85 ± 37 | 377 ± 187 | 58 ± 19 | 4 ± 2 | 260 ± 145 | 4427 ± 1623 | 513 ± 447 | 78.8 ± 17 | 1.46 ± 1.01 | 438 ± 174 | 14 ± 16 | 48 ± 14 | 4 ± 1 | 196 ± 79 | 167 ± 0 | 229 ± 8 |
| 12 | 69 ± 24 | 321 ± 153 | 41 ± 16 | 4 ± 2 | 188 ± 148 | 3244 ± 1467 | 525 ± 472 | 84.5 ± 16 | 1.44 ± 0.97 | 347 ± 169 | 10 ± 12 | 46 ± 10 | 4 ± 1 | 203 ± 78 | 171 ± 2 | 210 ± 14 |
| 13 | 85 ± 34 | 393 ± 141 | 53 ± 14 | 4 ± 1 | 224 ± 80 | 3765 ± 1055 | 535 ± 502 | 79.6 ± 20 | 2.71 ± 1.83 | 388 ± 110 | 11 ± 10 | 55 ± 0 | 4 ± 1 | 215 ± 44 | 175 ± 0 | 219 ± 17 |
| Mean ± SD | 97 ± 35 | 475 ± 207 | 58 ± 17 | 5 ± 2 | 290 ± 140 | 4180 ± 1337 | 516 ± 449 | 77.3 ± 21 | 1.59 ± 1.44 | 414 ± 139 | 10 ± 10 | 54 ± 8 | 4 ± 1 | 244 ± 70 | 279 ± 90 | 263 ± 48 |

*Frequency (n), Duration (minutes), RPE (AU), sRPE (AU), Player Load (AU), Tackles (n), Lower Body (LB) Repetitions (reps), Upper Body (UB) Repetitions (reps)

Descriptive observations of the weekly training load across the pre-season are presented as means ± SD in Figs 1–4 and described below.

## Changes in training load over preseason

**Between-week comparisons.** Week-to-week changes along with the between positional group variations between weeks are presented in Figs 1–4 for total distance (Fig 1A total weekly, Fig 2A mean sessional), high-speed meters (Fig 1B total weekly, Fig 2B mean sessional), average speed (Fig 2C), acceleration density (Fig 2D), player load (Fig 1C total weekly, Fig 2E mean sessional), tackles (Fig 1D total weekly, Fig 2F mean sessional), lower (Fig 3A total weekly, Fig 4A mean sessional) and upper body (Fig 3B total weekly, Fig 4B mean sessional) gym volume. Preseason block one (weeks 1–2) consisted of 16 sessions (9 field and 8 gym), block two (weeks 3–6) consisted of 29 sessions (14 field and 15 gym), block three (weeks 7–9) consisted of 10 sessions (6 field and 4 gym), and block 4 (weeks 10–13) consisted of 29 sessions (16 field and 13 gym). Field session intensity (i.e., acceleration density and average speed) increased over the 13 weeks. Upper body volume increased (Wk1. to Wk.13 diff. 150 repetitions, ES = 2.24 (1.69–2.79), very large, p < 0.001), while lower body volume (Wk.1 to Wk.13 diff. 762 repetitions, ES = 8.60 (8.06–9.15), very large, p < 0.001) and sRPE (Wk.1 to Wk.13 diff. 327.9 arbitrary units, ES = 2.04 (1.51–2.57), very large, p < 0.001) decreased across the pre-season.

**Within-week positional comparisons.** *Field-based training load*. In comparison to the forward playing group, backs covered significantly more high-speed meters in week two (503 m, ES = 0.89 (0.12–1.69), moderate, p = 0.024), and week 11 (479 m, ES = 0.84 (0.10–1.59), moderate, p = 0.028) (Fig 1B). There were no significant differences between positional groups for the mean sessions.

*Gym-based training load*. When compared to the forward playing group, backs completed more total upper body volume (backs: 391 ± 6 repetitions, forwards 369 ± 7 repetitions, ES = 0.33 (0.06–0.60), small, p = 0.025). Total weekly upper body volume was significantly different between playing positions in week three and week 12 (diff. 153 repetitions, ES = 2.28 (1.52–3.06), very large, p < 0.001; diff. 83 repetitions, ES = 1.24 (0.39–2.10), large, p < 0.005) (Fig 3B). Backs completed more lower body volume than forwards did in week one (diff. 82 repetitions, ES = 0.92 (0.15–17), moderate, p = 0.020) (Fig 3A).

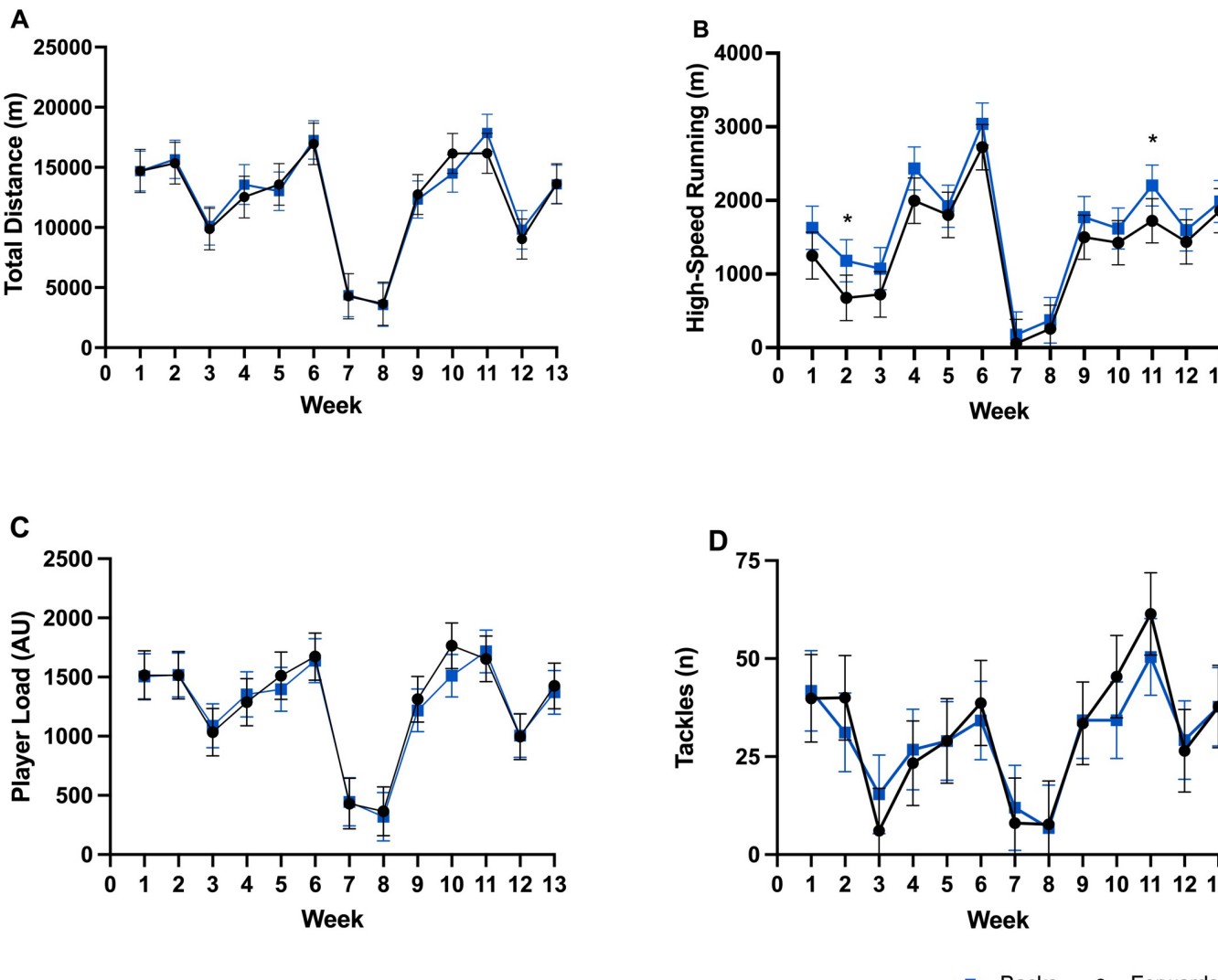

**Fig 1. Total weekly field-based external load, between positional groups, across weeks.** (A) Total distance, (B) High-speed running, (C) PlayerLoad, (D) Tackles. * Difference between forward and back playing positions ($p < 0.05$).

## Discussion

The purpose of this study was to quantify the field and resistance training load of academy rugby league during a pre-season period and compare training loads between forwards and backs. Findings revealed the average academy pre-season training session involved one hour of both gym and field training, with an associated sRPE of 244 ± 70 and 290 ± 140 AU, respectively. Within each of the gym sessions, players completed on average an estimated volume of 279 ± 90 repetitions of upper body and 263 ± 48 repetitions of lower body exercises. On field, players averaged 4180 ± 1337 m of total distance, 516 ± 449 m of high-speed running, at a rate of 77.3 ± 21 m·min$^{-1}$, while accumulating an acceleration density of 1.59 ± 1.44 m·s$^{-1}$, a PlayerLoad of 414 ± 139 AU, and 10 ± 10 tackles per session. This study is the first to report external and internal training load of both field and gym sessions in an academy rugby league pre-season period, comparing between positions.

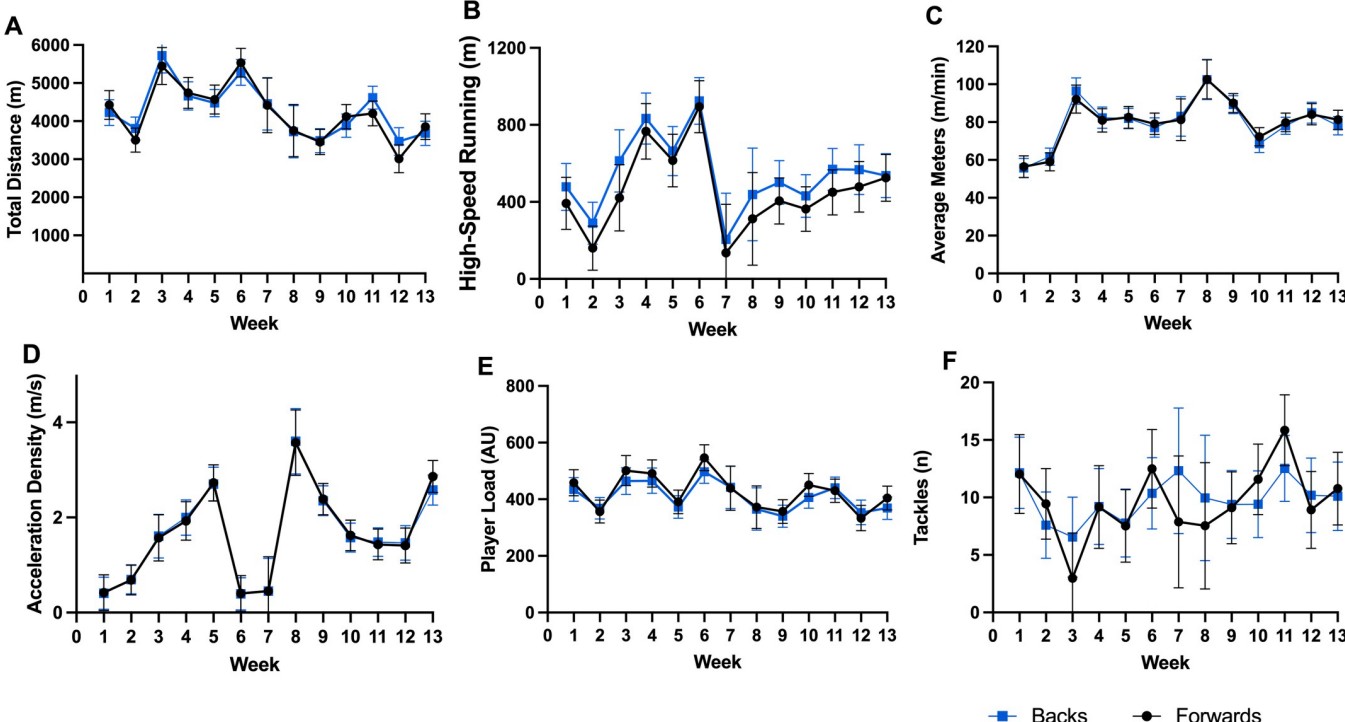

**Fig 2. Mean sessional field-based external load, between positional groups, across weeks.** (A) Total distance, (B) High-speed running, (C) Meters per minute average, (D) Acceleration density, (E) PlayerLoad, (F) Tackles.

Within a typical week during an academy rugby league pre-season period, players completed 6–9 training sessions with an equal frequency distribution between gym and field. Weekly training time comprised of 324 ± 137 minutes (3–8 hours) of training per week, which is lower than the previously reported 809 ± 224 minutes (10–17 hours) [26]. However, the values reported by McCormack et al. [26] should be interpreted with caution, as practitioners were asked to report an estimate of the frequency and duration of training. Whereas in the current study the frequency and duration reported can be interpreted confidently, as the reports were precisely captured via microtechnology units. The lower training volume compared to those reported by McCormack et al [26] may not come as a surprise, as U19 players are employed as professional athletes, but are part-time, often training while studying or working

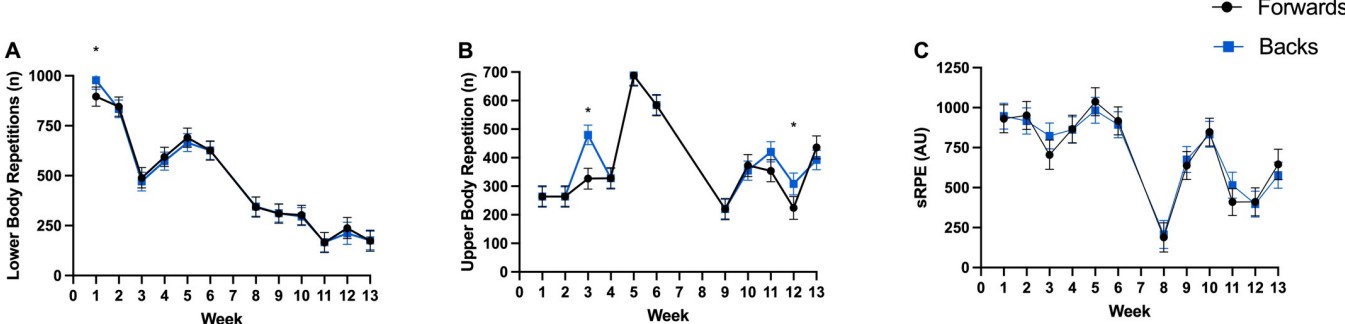

**Fig 3. Total weekly gym-based training load, between positional groups, across weeks.** (A) Lower body repetition volume, (B) Upper body repetition volume, (C) Session rating of percieved exertion. * Difference between forward and back playing positions (p < 0.05).

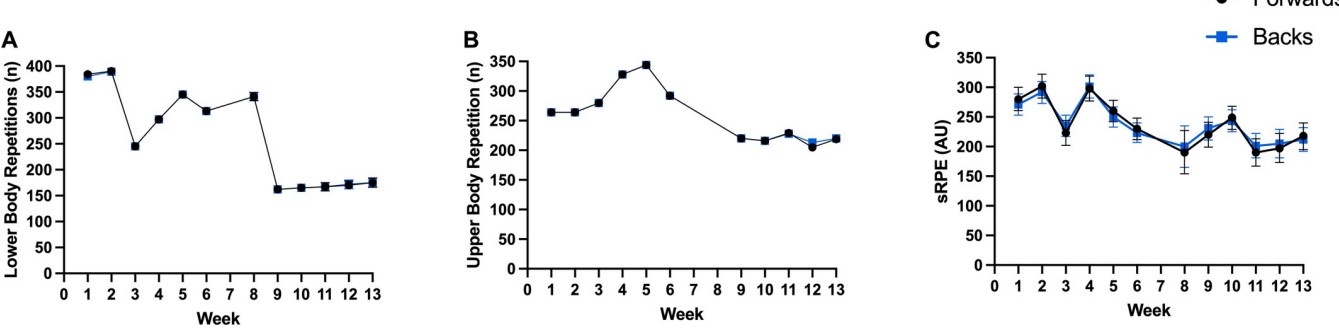

**Fig 4. Mean sessional gym-based training load, between positional groups, across weeks.** (A) Lower body repetition volume, (B) Upper body repetition volume, (C) Session rating of percieved exertion.

[23, 26, 27]. Yet they are still expected to follow high frequency training programmes ensuring they are adequately prepared for senior professional rugby league [36].

Rating of perceived exertion is one of the more commonly reported measure of internal training load. In the current study, RPE across the pre-season period was similar between field and gym sessions with a typical 4–5 AU rating, but still lower than values reported in senior professionals [10, 11]. The accumulated sRPE-TL reported for the current 13 week pre-season observation period (18,979 AU) is less than that previously reported in an academy RL preseason (22,670 AU) [25]. However, the sRPE-TL in the Dobbin (2018) study was accrued over the course of a 14-week period and a separate rating was given for conditioning which may explain the differences between studies. In the current study, overall training load decreased throughout the 13-weeks, this could have been a typical periodisation or match preparation strategy. Typically as the in-season approaches a decrease in volume and increase in intensity is expected [37], however this may contradict stage-appropriate training, where the emphasis should remain on enhancing physical qualities of academy athletes [23, 38, 39]. However, sRPE continues to be a valuable global measure of training load that has been validated in rugby league to help guide this process [10].

Improved precision of GPS and accelerometer technology has enabled the ability to capture detailed information on external training load [30] facilitating the development of specific training to prepare players for the rigours of competition. Total distance covered has been the most commonly reported locomotive variable, with players covering between 4–8 km during match play [20, 40]. In the current study, the academy players achieved these distances regularly throughout the preseason (average session distance 4180 ± 1337 m), however, the usefulness of total distance alone may be limited given the numerous ways (e.g., walking, jogging, sprinting) in which it can be accumulated [34]. Average speed (e.g., intensity of session) was higher per session in comparison to those reported in senior players (77.3 ± 21 vs. 72.6 ± 2.8 m·min$^{-1}$) [18, 29]. While the mean session duration (58 ± 17 min) and high-speed distance (516 ± 449 m) matched those reported by Black et al. [18], (51.9 ± 5 min and 496 ± 135 m, respectively). This is unsurprising as it has been previously reported that academy players are exposed to similar peak average running speeds to senior professionals, demonstrating the training stimulus at academy level is preparing athletes to progress to senior grade [41].

The participants in the current study averaged 10 ± 10 tackles per session, accumulating 30 ± 19 tackles per week. Tackling is a key contact event in RL, it has been recommended to quantify and monitor these events in training [42]. Hulin et al. [42] validated the ability of microtechnology devices to accurately count the frequency of tackles during RL matches, providing practitioners with a measure of contact load. In a meta-analysis characterizing the

physical demands of RL match-play, Glassbrook et al. [20] suggested that players typically experience 25.6 ± 4.3 collisions per game. Despite potential advantages of match-specific training, it may be unrealistic and undesirable for training to consistently replicate match demands. Therefore, the distribution of tackle exposures throughout the week, as was observed in the current study, serves as a viable strategy to prepare for competition. Training should function within an acceptable framework for athlete development, which considers volume, intensity, duration and frequency of training according to periodised plans and athlete needs [39, 43]. With a measure of tackle frequency readily available, this enables the practitioner to reflect on whether or not athletes are getting the necessary exposure at academy level in preparation for professional ranks. However, in the current study similar to field loads, there appeared to be a lack of periodisation to contact exposures over the duration of the pre-season (e.g., week 2 = 36 ± 23, week 3 = 12 ± 13). Improving our understanding of these events may improve training prescription [20].

Resistance training forms an important aspect of adolescent rugby player development, yet limited evidence exists exploring resistance training loads alongside field training [43, 44]. This study quantified gym-based training loads by providing details on volume (frequency, duration, repetitions), and subsequent internal load. The resistance training volume (3–4 sessions, averaging 54 ± 8 minutes) achieved in the current study is similar to the suggested 2–3 sessions per week that are deemed sufficient for the development of strength in adolescents [45]. An upper and lower-body split program was implemented and while upper body weekly loads remained constant throughout the 13-week pre-season period (383 ± 137 reps), lower body volume decreased. This could have been an attempt by practitioners to minimise fatigue as match preparation becomes a priority, and may not be the best strategy in the context of a long-term athlete development model. Resistance exercise however, is difficult to precisely quantify owing to its inherent complexity with numerous modifiable training variables contributing to the training dose [35]. Redman et al. [46] recently proposed monitoring both prescribed and actual resistance load of key exercises, providing a quantification of volume load (sets x reps x load) and training intensity (volume load/total reps). This is especially pertinent given reports from Weakley et al. [44] on the possible variation of adherence to prescribed practices in adolescent athletes. In summary, while providing an important starting point for the study of resistance training practices in academy rugby league, reports based on repetition volume leave out relevant details (i.e., load, tempo, intensity) that dictate the adaptive response.

The current findings showed limited positional differences for locomotor, tackle, and gym training loads, suggesting that training in academy rugby league may be homogenous between playing positions. In accordance with traditional periodisation models, training load must be varied to elicit optimal physiological adaptations [32]. Clear positional differences throughout match-play (e.g., backs engage in more high-speed sprinting, while forwards are involved in more collisions) [20, 24, 41] support the notion of position specific training approaches at the adolescent level, particularly because these positional differences become evident at older levels of competition [47]. Overall, training loads remained similar across all variables with the exception of HSR, whereby backs covered significantly more HSR distance in week two, four and eleven. Typically, the preseason period has an emphasis on general preparation, which may explain the uniformity in training loads between forwards and backs, where individual and specific preparation is further emphasised during the in-season. However reports from Thornton et al. [38] suggest positional-specific prescription may not be necessary within elite youth RL athletes purely from a physical perspective (i.e. running intensity and/or speed). Practitioners may therefore wish to manipulate drills between age and positional groups to appropriately reflect technical and tactical abilities [38]. Improving our understanding of the

demands of rugby league (training and match-play) and of how the demands may differ across playing positions should further improve training design [20]. More research is required to determine positional demands and appropriate training approaches at the adolescent level [47].

## Limitations

While this study provides useful and practical data, it is important to consider these are the training loads of academy athletes belonging to one professional academy and may not be representative of all other rugby league academy training. Secondly, while the current study has attempted to improve upon the accuracy of reporting, training volume comparisons should be interpreted with caution because of methodological differences in volume calculations. While the current study aimed to consider the multi-modal (field and gym) aspects of training, detailing training volumes by physical (i.e., resistance training, conditioning, speed) and rugby content (i.e., skills, tactics, combat) would capture a more representative sample of the concurrent training practices within academy rugby league. Additionally, quantifying resistance exercise remains a challenge, numerous independent variables that elicit adaptation were unable to be captured (e.g. type of exercise, relative load lifted, inter-set rest periods, and repetition velocity) [35]. While the current study reported on repetition volume, the addition of sRPE attempts to capture the "load" associated with gym related content. Advancements in technology and data management enable the ability to capture the different constructs that influence performance outcomes which future studies may wish to explore [48, 49]. Currently, the available training data for rugby league coaches at any level remains limited, therefore match-play data is often used as the reference for what may be deemed appropriate [18, 43].

## Practical applications

While coaches may use established match-play demands as a benchmark to facilitate the development of future training programmes, this study highlights the usefulness in paying particular attention to training data. Training should function within an acceptable framework for athlete development, which considers volume, intensity, duration, and frequency of training according to periodised phases and athlete needs [43]. This study provides an example of an evaluation of a pre-season training loads that could be replicated by practitioners in evaluating training. In order to maximise physiological changes during a preseason, it is vital to understand the day-to-day training of players [50]. Quantifying the training loads of academy rugby league serves as a critical first step to maximising their athletic preparation. By quantifying the demands of training there is scope to develop specific training drills to more appropriately prepare players for the rigours of competition [17]. This study reinforces that coaches and sport scientists should work closely to use the available external load data (GPS) alongside measures of internal load (sRPE) to adopt a more scientific approach to training prescription (e.g., load management, periodisation) [5]. Having these data readily available enables the practitioner to plan, deliver, and evaluate whether or not they are achieving desired targets, while gaining an understanding of the dose-response relationship [48, 51, 52]. Furthermore, translating training load data into meaningful information can facilitate performance discussions accounting for the data while not solely relying on opinion [53].

## Conclusion

In conclusion, this study has quantified the field- and resistance-based training loads of an academy rugby league preseason, with consideration to playing position and the tackle. Findings showed that training load is distributed throughout the training week as it is unfeasible

for training to reflect the demands of match-play during each and every session, particularly in contact sports such as rugby league [54]. However, the results highlight uniformity of training loads across playing positions, and unclear progressions of load throughout the preseason period. It is unlikely that a one-size-fits all approach to training adequately prepares players across a range of playing positions for the specific contact demands and movement patterns experienced in match-play [55]. Therefore, practitioners and researchers alike are encouraged to continue quantifying training load to inform future planning, delivery, and evaluation. Future research considering training characteristics of rugby league should explore how to maximise the multi-modal training practices adopted in these settings and whether or not specific playing position match-demands are tailored for.

## Acknowledgments

This research was supported by the Leeds Rhinos, we would like to thank them for their support in providing participant data. We would also like to acknowledge the practitioners who took their time to deliver the training and facilitate the data collection.

## Author Contributions

**Conceptualization:** David Anthony Moore, Ben Jones, Jonathon Weakley, Kevin Till.

**Data curation:** Sarah Whitehead.

**Formal analysis:** David Anthony Moore.

**Investigation:** David Anthony Moore.

**Methodology:** David Anthony Moore.

**Supervision:** Ben Jones, Jonathon Weakley, Kevin Till.

**Visualization:** David Anthony Moore.

**Writing – original draft:** David Anthony Moore.

**Writing – review & editing:** Ben Jones, Jonathon Weakley, Sarah Whitehead, Kevin Till.

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
