## [Decision Letter · Decision Letter 0]

10 May 2022

PONE-D-22-08332The Field and Resistance Training Loads of Academy Rugby League Players during a Pre-Season: Comparisons Across Playing PositionsPLOS ONE

Dear Dr. Moore,

Thank you for submitting your manuscript to PLOS ONE. After careful consideration, we feel that it has merit but does not fully meet PLOS ONE’s publication criteria as it currently stands. Therefore, we invite you to submit a revised version of the manuscript that addresses the points raised during the review process.

ACADEMIC EDITOR:Dear Authors, your ms has been revised by two experts in the field that advise some minor changes.

We look forward to receiving your revised manuscript.

Kind regards,

Emiliano Cè

Academic Editor

PLOS ONE

Journal Requirements:

Reviewers' comments:

Reviewer's Responses to Questions

**Comments to the Author**

1. Is the manuscript technically sound, and do the data support the conclusions?

Reviewer #1: Yes

Reviewer #2: Yes

2. Has the statistical analysis been performed appropriately and rigorously? 

Reviewer #1: Yes

Reviewer #2: Yes

3. Have the authors made all data underlying the findings in their manuscript fully available?

Reviewer #1: Yes

Reviewer #2: Yes

4. Is the manuscript presented in an intelligible fashion and written in standard English?

Reviewer #1: Yes

Reviewer #2: Yes

5. Review Comments to the Author

Reviewer #1: Dear Authors,

I appreciate the opportunity to comment to the authors on their manuscript titled "The Field and Resistance Training Loads of Academy Rugby League Players during a Pre-Season: Comparisons Across Playing Positions". The manuscript's issue is interesting and relevant in the current context of knowledge at the level of youth training in Rugby. I consider that the relevance of the study lies in the limited research available that explores the training load of academy rugby league players.

The purpose of the study was to quantify the field and resistance training loads of academy rugby league players during a pre-season period and compare training loads between playing positions (i.e., forwards vs. backs).

The study is well structured, with the results adequately discussed.

I do however have some concerns that I point out to below:

Procedures section,

# Please indicate the power sample calculation. Power calculations allow the reader to know how many subjects are required in order to avoid a type I or a type II error.

# line 137-138 - I am of the opinion that it is more appropriate to cite for the determination of the s-RPE the reference of :

Foster, C, Florhaugh, JA, Franklin, J, Gottschall, L, Hrovatin, LA, Parker, S, Doleshai, P, Dodge, C. A new approach to monitoring exercise training. J Strength Cond Res 15: 109-115, 20 2001.

# line 157 and remaining text - please correct the units of the speed threshold (m/s) and also the average speed (m/min)

# A comment for future studies - the repetition method is a very simple method for estimating training volume but it offers a poor estimate of the amount of work or volume of training completed in a resistance training bout or training programme. A more accurate approach would be to either directly quantify work accomplished, or use estimates which account for the actual weight lifted by the athlete when attempting to either quantify or equate training loads when comparing various training interventions.

# Please clarify why the authors did not use the acceleration density index metric that allows quantifying the intensity of speed change activity exclusively during locomotive work, rather than the whole time period. In my opinion this might be preferred to acceleration density as a way to analyze speed change demands excluding rest periods.

#line 163, 332 and 357- for confirming whether it is actually the reference [31]Varley et al (2012) that the authors intend to cite, since it does not address the repetition method.

#please standardize throughout the text the way in which the p-value is expressed, e.g., 0.00 or .00

# In the results section, I suggest indicating the size effects of the comparisons made and with statistically significant differences.

# I suggest that the figures could be improved in terms of their sharpness.

# In figure 4 (A) Lower Body Repetitions volume, are the data from the two positional groups presented?

References

# Reference [55] is not cited in the text:

Tee JC, Lambert MI, Coopoo Y. GPS comparison of training activities and game demands of professional rugby union. Int J Sport Sci Coach. 2016;11(2):200–11

BW

João Paulo Brito

Reviewer #2: The paper is well written and provides a clear and accurate representation of the training demands of adolescent rugby players. I recommend this paper for publication.

Prior to publication, the authors should provide the specific URL details where the data set can be accessed.

6. PLOS authors have the option to publish the peer review history of their article (what does this mean?). If published, this will include your full peer review and any attached files.

Reviewer #1: **Yes: **João Paulo Brito

Reviewer #2: No

---

## [Author Response · Author response to Decision Letter 0]

28 Jun 2022

Thank you for your recent review of the submitted article 'The field and resistance training loads of academy rugby league players during a preseason: Comparisons across playing positions'. We have addressed the comments and provided details in the cover letter. We believe the comments were fair, helpful and allowed us to improve a number of areas in the paper. We have also addressed the editing requirements of the journal and provided repository information.

---

## [Decision Letter · Decision Letter 1]

27 Jul 2022

The field and resistance training loads of academy rugby league players during a pre-season: Comparisons across playing positions

PONE-D-22-08332R1

Dear Dr. Moore,

We’re pleased to inform you that your manuscript has been judged scientifically suitable for publication and will be formally accepted for publication once it meets all outstanding technical requirements.

Kind regards,

Emiliano Cè

Academic Editor

PLOS ONE

Additional Editor Comments (optional):

Reviewers' comments:

Reviewer's Responses to Questions

**Comments to the Author**

1. If the authors have adequately addressed your comments raised in a previous round of review and you feel that this manuscript is now acceptable for publication, you may indicate that here to bypass the “Comments to the Author” section, enter your conflict of interest statement in the “Confidential to Editor” section, and submit your "Accept" recommendation.

Reviewer #1: All comments have been addressed

Reviewer #2: All comments have been addressed

2. Is the manuscript technically sound, and do the data support the conclusions?

Reviewer #1: Yes

Reviewer #2: Yes

3. Has the statistical analysis been performed appropriately and rigorously? 

Reviewer #1: Yes

Reviewer #2: Yes

4. Have the authors made all data underlying the findings in their manuscript fully available?

Reviewer #1: Yes

Reviewer #2: Yes

5. Is the manuscript presented in an intelligible fashion and written in standard English?

Reviewer #1: Yes

Reviewer #2: Yes

6. Review Comments to the Author

Reviewer #1: Dear Authors,

After reviewing the manuscript I found that the authors took into account all concerns addressed in the first review.

From the reported appreciation I am of the opinion that it is now suitable for publication.

Congratulations on the work done.

Reviewer #2: Congratulations! This is a useful contribution to the literature

7. PLOS authors have the option to publish the peer review history of their article (what does this mean?). If published, this will include your full peer review and any attached files.

Reviewer #1: **Yes: **João Paulo Brito

Reviewer #2: No

---

## [Editor Report · Acceptance letter]

1 Aug 2022

PONE-D-22-08332R1 

The field and resistance training loads of academy rugby league players during a pre-season: Comparisons across playing positions 

Dear Dr. Moore:

I'm pleased to inform you that your manuscript has been deemed suitable for publication in PLOS ONE. Congratulations! Your manuscript is now with our production department. 

Kind regards, 

on behalf of

Professor Emiliano Cè 

Academic Editor

PLOS ONE